# Prescription of concomitant medications in patients treated with Nifurtimox Eflornithine Combination Therapy (NECT) for *T.b. gambiense* second stage sleeping sickness in the Democratic Republic of the Congo

**Andrea Kuemmerle**[1,2], **Caecilia Schmid**[1,2¤], **Victor Kande**[3], **Wilfried Mutombo**[3,4], **Medard Ilunga**[3], **Ismael Lumpungu**[3], **Sylvain Mutanda**[3], **Pathou Nganzobo**[3], **Digas Ngolo**[3,4], **Mays Kisala**[5], **Olaf Valverde Mordt**[4]*

**1** Swiss Tropical and Public Health Institute, Basel, Switzerland, **2** University of Basel, Basel, Switzerland, **3** Programme National de Lutte contre la Trypanosomiase Humaine Africaine, Democratic Republic of the Congo, **4** DND*i*, Geneva, Switzerland, **5** Bureau Diocesain d'Oeuvres Medicales, Kikwit, Democratic Republic of the Congo

¤ Current address: Swiss Clinical Trial Organisation, Basel, Switzerland
* ovalverde@dndi.org

**Data Availability Statement:** Data cannot be shared publicly as written permission to publicly

## Abstract

### Background

Nifurtimox eflornithine combination therapy (NECT) to treat human African trypanosomiasis (HAT), commonly called sleeping sickness, was added to the World Health Organisation's (WHO) Essential Medicines List in 2009 and to the Paediatric List in 2012. NECT was further tested and documented in a phase IIIb clinical trial in the Democratic Republic of Congo (DRC) assessing the safety, effectiveness, and feasibility of implementation under field conditions (NECT-FIELD study). This trial brought a unique possibility to examine concomitant drug management.

### Methodology/Principal findings

This is a secondary analysis of the NECT-FIELD study where 629 second stage gambiense HAT patients were treated with NECT, including children and pregnant and breastfeeding women in six general reference hospitals located in two provinces. Concomitant drugs were prescribed by the local investigators as needed. Patients underwent daily evaluations, including vital signs, physical examination, and adverse event monitoring. Concomitant medication was documented from admission to discharge. Patients' clinical profiles on admission and safety profile during specific HAT treatment were similar to previously published reports. Prescribed concomitant medications administered during the hospitalization period, before, during, and immediately after NECT treatment, were mainly analgesics/antipyretics, anthelmintics, antimalarials, antiemetics, and sedatives. Use of antibiotics was reasonable and antibiotics were often prescribed to treat cellulitis and respiratory tract infections. Prevention and treatment of neurological conditions such as convulsions, loss of consciousness, and coma was used in approximately 5% of patients.

share personal medical information was not received from the enrolled subjects. Data can be made available to qualified researchers upon request after signing a confidentiality agreement. Data requests may be sent to the Department of Medicine, Swiss Tropical and Public Health Institute (medres@swisstph.ch).

**Funding:** This secondary analysis was funded by DNDi. The NECT-FIELD study was sponsored by UK aid, UK; the Swiss Agency for Development and Cooperation (SDC), Switzerland; Médecins Sans Frontières (MSF) International and other private foundations. The NECT-FIELD study funders had no role in study design, analysis, decision to publish, or preparation of the manuscript.

**Competing interests:** The authors have declared that no competing interests exist.

## Conclusions/Significance

The prescription of concomitant treatments was coherent with the clinical and safety profile of the patients. However, some prescription habits would need to be adapted in the future to the evolving available pharmacopoeia. A list of minimal essential medication that should be available at no cost to patients in treatment wards is proposed to help the different actors to plan, manage, and adequately fund drug supplies for advanced HAT infected patients.

## Trial registration number

The initial study was registered at ClinicalTrials.gov, number NCT00906880.

## Author summary

Sleeping sickness is a neglected tropical disease caused by a parasite, the trypanosome, transmitted by the tse-tse fly. It affects people in Sub-Saharan Africa, most of them living in poor rural settings with limited access to healthcare. If the disease remains untreated, it usually progresses from a haemo-lymphatic invasion into a second stage or neuro-encephalitic infection evolving into coma and death. When these second stage patients are referred to treatment centres, the disease is often advanced. Therefore, on top of sleeping sickness treatment, additional care and medications are regularly required. We report here the concomitant medications that were prescribed to second stage sleeping sickness patients, including vulnerable populations such as children, and pregnant and breastfeeding women, in the context of a phase IIIb trial assessing the safety, effectiveness, and feasibility of implementation of the Nifurtimox Eflornithine Combination Therapy (NECT) in field conditions. Our objective is to provide evidence to national health systems or private actors to plan and fund their essential medicine supplies adapted to second stage sleeping sickness patients.

## Introduction

Human African trypanosomiasis (HAT), also called sleeping sickness, affects people in sub-Saharan Africa who often live in remote or insecure areas with limited access to healthcare. The disease is caused by a trypanosome infection after a bite from a tse-tse fly and evolves from a haemo-lymphatic invasion (first stage) into a neuro-encephalitic infection (second stage) that leads to severe sleep disturbances, neurological and psychiatric disorders, coma, and in most cases to death if untreated [1]. Until 2009, treatment for the second stage *T.b. gambiense* HAT was limited to melarsoprol, an arsenic derivative with high toxicity [2] and in some areas, reduced efficacy [3–5], or eflornithine monotherapy, a burdensome treatment consisting in 56 slow infusions over 2 weeks [6].

Nifurtimox Eflornithine Combination Therapy (NECT) was added to the World Health Organization's Essential Medicines List (WHO EML) in 2009 [7] and to the Paediatric WHO EML in 2012 [8]. Since then, a new oral molecule, fexinidazole, has been further assessed and was granted a positive opinion from the European Medicine Agency (EMA) for the treatment of first and second stage HAT in 2018 [9, 10]. Following the introduction of fexinidazole, NECT will still remain the first choice treatment in patients presenting a clinical picture consistent with severe second stage HAT with $\geq$ 100 white blood cell (WBC)/μL in the cerebrospinal fluid (CSF), or those unable to eat. It also remains the first choice for pediatric patients

younger than 6 years old or weighting less than 20 kg presenting with > 5 WBC/μL or trypanosomes in the CSF [11, 12].

From 2009 to 2012, NECT was tested and documented in a phase IIIb clinical trial (NECT-FIELD study) in the Democratic Republic of Congo (DRC) [13]. In the NECT-FIELD study, the safety, effectiveness, and feasibility of implementation under field conditions were assessed. The in-hospital safety profile of NECT administered to a wide population, including vulnerable people such as children, and pregnant and breastfeeding women, was already published by Schmid et al [13], while the 24-months effectiveness analysis is under preparation and will be submitted for publication.

Before and during the HAT specific treatment the patients received an array of concomitant medication to treat co-morbidities or manage adverse events. The NECT-FIELD study brought a unique opportunity to examine in-hospital patient management, given concomitant diseases and adverse events in second stage HAT patients.

The present analysis describes the concomitant medication prescribed during hospitalization before and during treatment with NECT. In our knowledge, this has never been done so far in any clinical trial involving second stage HAT patients. The main objective of this work is to help national health systems or private actors to plan, develop, and update the essential medicine supplies adapted to their second stage HAT gambiense patients including vulnerable populations such as children as well as pregnant and breastfeeding women.

## Methods

### Design and study population

This is a secondary analysis of the NECT-FIELD study conducted between 2009 and 2012 in six HAT treatment centres located in reference district hospitals in the DRC and included 629 patients treated with NECT. The design, methods, and in-hospital safety results of the NECT-FIELD study are described in the published report by Schmid et al [13]. Briefly, all second stage HAT patients admitted to the treatment centres received NECT, the co-administration of nifurtimox (oral 15 mg/kg/day, three times a day) for 10 days and eflornithine (slow intravenous infusions, 400 mg/kg/day, twice a day) for 7 days. All patients were hospitalised during the entire treatment period.

Prior to NECT, the patients received standard pre-treatment according to the national guidelines, consisting of antimalarial, anthelminthic, and, if required, antipyretic/analgesic medication. Concomitant medication was prescribed according to the decision of the investigator and the local guidelines. An emergency pharmacy for eventual life-threatening issues was provided by the study coordination to the Investigators while the other drugs were routinely supplied by the national HAT program or the hospital pharmacy.

### Collected data

Patients underwent daily evaluations, including vital signs, physical examination, and adverse event (AE) monitoring. The concomitant medications were documented and details such as the reason for use, dates, dosages, and route and duration of administration were recorded. AEs and their severity were reported from the first day of treatment until discharge from hospital and were assessed along with their severity and possible relation to the study treatment [14]. The AEs and concomitant medications were coded with the MedDRA dictionary (version 11) and the WHO-Drug dictionary (version Q2.06) respectively. AEs observed in ≥ 5% of the patients and concomitant medications prescribed in ≥ 2.5% have been included in the result tables for the assessment of concomitant medication prescriptions. AEs and concomitant drugs

that appeared with lower frequencies but were of medical relevance were included after discussion between AK and OVM.

## Statistics

The main analysis set is the intention to treat (ITT) population, which includes all subjects who received at least one dose of study drug. Statistical evaluations are descriptive. Means, standard deviations, and number of patients are provided for continuous variables, as well as frequency distributions for binary and categorical variables. Statistical analyses were performed using the SAS software version 9.1 (SAS Institute, Cary, NC) and Epi Info 7.0.

## Ethical considerations

The NECT-FIELD study received approval from two Ethics Committees: the Ethics Committee of both cantons of Basel (EKBB, Basel, Switzerland; 26 February, 2009) and the local Ethics Committee in the DRC (Comite d'ethique sur la Trypanosomiase Humaine Africaine, Kinshasa, Democratic Republic of the Congo, 7 May 2009). Eligible patients provided written informed consent before enrolment into the study. In case of minors, severely ill, or mentally impaired patients unable to fully consent, written informed consent was obtained from her/his parent(s)/guardian(s). Whenever possible, depending on age and level of understanding, the children received the information and their assent was obtained.

# Results

## Study population and clinical profile on admission

Of the 629 patients included in this analysis, 100 were children below 12 years of age, 33 were breastfeeding women, and 14 were pregnant women (Table 1). The patient flow is published

**Table 1. Demographic characteristics of the study population.**

| Characteristics | Number of patients treated n (%)* |
|---|---|
| Total number of patients (N) | 629 |
| Male/female ratio | 1.3 |
| Children 0–11 years | 100 (15.9) |
| Adolescents/adults > 11 years | 529 (84.1) |
| Breastfeeding women | 33 (5.2) |
| Pregnant women | 14 (2.2) |
| Age, mean (SD) years# | 30 (16.2) |
| Median | 28 |
| Range (min-max) | 1–77 |
| Study sites | |
| Kasai Oriental Province | |
| Dipumba | 146 (23.2) |
| Katanda | 132 (21.0) |
| Ngandajika | 94 (14.9) |
| Bandundu Province | |
| Bandundu | 98 (15.6) |
| Kwamouth | 97 (15.4) |
| Yasa Bonga | 62 (9.9) |

*Unless otherwise specified

#age unknown for 2 patients

in the in-hospital safety report [13]. HAT signs and symptoms reported by the patients on admission are shown in Table 2.

## Treatment emergent adverse events

The reporting of treatment emergent AEs showed that above all, patients developed gastrointestinal disorders including nausea and vomiting, general disorders like fever, and neurological and psychiatric disorders like asthenia, headaches, dizziness, convulsions, insomnia, and agitation that are difficult to distinguish from the disease itself (Table 3). Less frequently but of medical importance, potentially severe or life-threatening AEs if not treated such as coma, loss of consciousness, specific psychiatric disorders, infections, and cardiac disorders were also observed.

## Prescription of concomitant medication

Medication given before study treatment started consisted mainly in anti-parasitic pre-treatment according to HAT national guidelines, usually 1–3 days before the start of the first dose of NECT and lasting normally 3 days. This anti-parasitic pre-treatment usually comprised an antimalarial and an anthelmintic (Table 4). Depending on the treatment centre, paracetamol was either routinely administered after the lumbar puncture or when the patients complained of headaches. The majority of the patients received the antimalarial pre-treatment sulfadoxine/ pyrimethamine combination, with the exception of one centre, where patients received artesunate. Severe malaria cases were managed at all sites with intravenous isotonic quinine infusion. Concomitant medication during NECT was common in all study populations and most patients received at least one concomitant medication (Table 4). Administered treatments were mainly analgesics and antipyretics, anthelmintics, antiemetics, and sedatives.

The most frequently administered antibiotics during NECT were amoxicillin, ampicillin, ciprofloxacin, and gentamicin and were mainly prescribed to treat injection site infections (including cellulitis) or respiratory tract infections.

## Discussion

We report an evaluation of the concomitant medication prescribed to the 629 patients included in the NECT-FIELD study conducted at six HAT treatment centres in the DRC [13].

**Table 2. HAT signs and symptoms reported by the patient on admission.**

| Percentage of patients with (%)* | Children 0–11 years | Adolescents & adults >11 years | Breast-feeding women | Pregnant women | All patients |
|---|---|---|---|---|---|
| | N = 100 | N = 482‡ | N = 33 | N = 14 | N = 629 |
| Sleeping disorders | 80.0 | 80.5 | 72.7 | 64.3 | 79.7 |
| Headache | 50.0 | 78.6 | 84.8 | 92.9 | 74.7 |
| Asthenia | 44.0 | 64.5 | 54.5 | 92.9 | 61.4 |
| Fever history | 77.0 | 56.0 | 42.4 | 71.4 | 59.0 |
| Pruritus | 44.0 | 61.8 | 57.6 | 35.7 | 58.2 |
| Weight loss | 46.0 | 55.8 | 54.5 | 78.6 | 54.7 |
| Tremor | 35.0 | 45.9 | 30.3 | 57.1 | 43.6 |
| Walking disorder | 28.0 | 45.0 | 15.2 | 57.1 | 41.0 |
| Behavioural disorder | 42.0 | 32.8 | 24.2 | 21.4 | 33.5 |
| Convulsions | 15.0 | 3.5 | 3.0 | 7.1 | 5.4 |

*Unless otherwise specified

‡all adolescents and adults except breastfeeding / pregnant women

**Table 3. Treatment emergent adverse events (AEs).**

| Percentage of patients with (%)* | Children 0–11 years | Adolescents & adults >11 years | Breast-feeding women | Pregnant women | All patients |
|---|---|---|---|---|---|
| | N = 100 | N = 482[‡] | N = 33 | N = 14 | N = 629 |
| Treatment emergent adverse events (AEs) | | | | | |
| Patients with at least 1 adverse event | 92.0 | 92.1 | 84.8 | 100.0 | 91.9 |
| Death | 0 | 1.9 | 0 | 7.1 | 1.6 |
| Gastrointestinal disorders | 43.0 | 63.9 | 66.7 | 92.9 | 61.4 |
| Vomiting | 31.0 | 43.6 | 57.6 | 78.6 | 43.1 |
| Nausea | 13.0 | 22.0 | 6.1 | 21.4 | 19.7 |
| Colitis or diarrhoea | 13.0 | 13.3 | 6.1 | 7.1 | 12.7 |
| Pain and discomfort of the gastrointestinal tract[%] | 5.0 | 16.2 | 18.2 | 35.7 | 14.9 |
| General disorders and administration site condition | 57.0 | 41.7 | 60.6 | 85.7 | 46.1 |
| Fever | 44.0 | 25.9 | 42.4 | 28.6 | 29.7 |
| Asthenia | 13.0 | 16.8 | 27.3 | 57.1 | 17.6 |
| Injection site disorders[&] | 7.0 | 4.8 | 6.1 | 7.1 | 5.2 |
| Nervous system disorders | 21.0 | 37.1 | 24.2 | 57.1 | 34.3 |
| Headache | 8.0 | 15.4 | 12.1 | 35.7 | 14.5 |
| Dizziness | 0 | 13.1 | 6.1 | 14.3 | 10.7 |
| Convulsions | 10.0 | 9.1 | 9.1 | 0 | 9.1 |
| Tremor | 4.0 | 3.3 | 0 | 0 | 3.2 |
| Coma or loss of consciousness | 1.0 | 1.7 | 3.0 | 7.1 | 1.7 |
| Metabolism and nutrition disorders | 22.0 | 27.8 | 12.1 | 28.6 | 26.1 |
| Anorexia | 21.0 | 27.2 | 9.1 | 28.6 | 25.3 |
| Psychiatric disorders | 9.0 | 17.6 | 12.1 | 0 | 17.6 |
| Insomnia | 3.0 | 7.7 | 0 | 0 | 6.4 |
| Agitation | 5.0 | 5.6 | 12.1 | 0 | 5.7 |
| Behavioural disorders | 0 | 1.9 | 0 | 0 | 1.4 |
| Mood disorders | 1.0 | 1.0 | 3.0 | 0 | 1.1 |
| Mental confusion | 1.0 | 0.8 | 0 | 0 | 0.8 |
| Hallucinations or delirium | 0 | 1.5 | 0 | 0 | 1.1 |
| Psychosis or acute psychosis | 0 | 0.2 | 3.0 | 0 | 0.3 |
| Depression | 0 | 0.2 | 0 | 0 | 0.2 |
| Musculoskeletal and connective tissue disorders | 4.0 | 15.4 | 15.2 | 21.4 | 13.7 |
| Lumbago | 2.0 | 6.2 | 6.1 | 21.4 | 5.9 |
| Neck pain | 0 | 3.9 | 3.0 | 0 | 3.2 |
| Skin and sub-cutaneous tissue disorders | 9.0 | 9.8 | 6.1 | 7.1 | 9.4 |
| Pruritus or cutaneous pruritus | 7.0 | 5.8 | 3.0 | 0 | 5.1 |
| Dermatitis or cutaneous eruption | 2.0 | 1.0 | 0 | 0 | 1.1 |
| Vascular disorders | 3.0 | 8.3 | 6.1 | 14.3 | 7.5 |
| Hypotension | 3.0 | 4.8 | 6.1 | 14.3 | 4.8 |
| Hypertension | 0 | 2.9 | 0 | 0 | 2.2 |
| Cardiac disorders | 4.0 | 7.9 | 3.0 | 7.1 | 7.0 |
| Palpitation or arrhythmia | 3.0 | 5.8 | 3.0 | 0 | 5.1 |
| Bradycardia | 0 | 1.2 | 0 | 0 | 1.0 |
| Tachycardia | 1.0 | 0.8 | 0 | 7.1 | 1.0 |
| Infections and infestations | 8.0 | 4.6 | 3.0 | 14.3 | 5.2 |
| Respiratory infections[$] | 2.0 | 1.0 | 7.1 | 0 | 1.3 |
| Cellulitis | 1.0 | 1.0 | 0 | 7.1 | 1.1 |

(*Continued*)

**Table 3.** (Continued)

| Percentage of patients with (%)* | Children 0–11 years | Adolescents & adults >11 years | Breast-feeding women | Pregnant women | All patients |
|---|---|---|---|---|---|
| | N = 100 | N = 482‡ | N = 33 | N = 14 | N = 629 |
| Malaria | 1.0 | 0.6 | 3.0 | 0 | 0.6 |
| Septic shock | 0 | 0.2 | 0 | 7.1 | 0.3 |
| Blood and lymphatic system disorders | 3.0 | 1.2 | 0 | 7.1 | 1.6 |
| Anaemia | 2.0 | 1.0 | 0 | 7.1 | 1.3 |

*Unless otherwise specified

‡all adolescents and adults except breastfeeding / pregnant women

%the following AEs have been grouped under Pain and discomfort of the gastrointestinal tract: heartburn, lower abdominal pain, left hypochondrium pain, abdominal pain, dyspepsia, dysphagia, epigastralgia, upset stomach

& the following AEs have been grouped under injection site disorders: pain at the injection site, pain at the catheter site, extravasation and injection site reaction.

$including pneumonia and bronchopneumonia

The patients' clinical and safety profiles were characteristic of second stage HAT patients [15–19]. It therefore appears that drugs were not only used to treat the treatment emergent AEs but also the symptoms related to the disease itself.

The known bone marrow toxicity of eflornithine [6] and the repeated eflornithine perfusions raised some concerns about the increased risk of bacterial infections in the field and the consequent use of antibiotics. The proportion of treated patients with antibiotics remained reasonable in our context which is reassuring.

Another concern in second stage patients are the neurological complications. Dexamethasone and Ringer lactate, administered intravenously in 5.2% and 2.7% of the patients, are drugs that are used in this setting to treat or prevent convulsions and coma. It seems that they have been administered at an appropriate frequency.

The concomitant drugs were prescribed in coherence with the clinical and safety profile of the study population even if some prescription habits would need to be improved. For example, one centre provided artesunate monotherapy as standard pre-treatment which is suboptimal considering the general recommendation of using it in combination. It also appeared that some classical field prescription practices such as the utilisation of metamizol and papaverine, administered in 25.4% and 2.5% of the patients respectively might be considered as suboptimal at first sight. However, after discussion with the investigators these choices were either driven by the availability of drugs at the centres or by some medical decisions. At the time of the study, between 2008 and 2010, metamizol, due to the suspicion of bone marrow toxicity, was only recommended when other antipyretics were inefficient or when the fever needed to be brought down urgently, above all in children. In this specific population, metamizol was often preferred because the intravenous route made its administration easier. Neverthelesss, metamizol has been banned in DRC since and treatment with NSAID such as ibuprofen or diclofenac would therefore be recommended today. Papaverine was administered preferably to pregnant women to treat gastrointestinal disorders such as nausea and vomiting, because metoclopramide is contra-indicated in this population.

In a low-resource setting, drug supply choices of the centers may partially be driven by the scarce financial resources but also by their availability at the hospital and surrounding or if the drug was supplied by the national HAT program. For example, the choice to prescribe levamisole and cotrimoxazole was mainly driven by the fact that these drugs were supplied by the HAT national program and their availability. However, data on concomitant drug supply and

**Table 4. Prescription of concomitant medication.**

| Percentage of patients treated with (%) | Children 0–11 years | Adolescents & adults > 11 years | Breast-feeding women | Pregnant women | All patients |
|---|---|---|---|---|---|
| | N = 100 | N = 482* | N = 33 | N = 14 | N = 629 |
| (A) BEFORE HAT TREATMENT | | | | | |
| Patients with at least 1 treatment | 89.0 | 85.3 | 72.7 | 85.7 | 85.2 |
| Anthelmintics | | | | | |
| Mebendazole | 33.0 | 32.8 | 27.3 | 28.6 | 32.4 |
| Antiprotozoals | | | | | |
| Sulfadoxine/pyrimethymine | 73.0 | 76.1 | 60.6 | 85.7 | 75.0 |
| Artesunate | 7.0 | 6.8 | 12.1 | 0 | 7.0 |
| Quinine | 8.0 | 1.7 | 0 | 0 | 2.5 |
| Analgesics, antipyretics | | | | | |
| Paracetamol | 5.0 | 9.1 | 12.1 | 7.1 | 8.6 |
| (B) DURING NECT TREATMENT | | | | | |
| Patients with at least 1 treatment | 90.0 | 95.4 | 97.0 | 100.0 | 94.8 |
| Antibacterials | | | | | |
| Amoxicillin | 8.0 | 5.8 | 3.0 | 7.1 | 6.0 |
| Sulfamethoxazole/trimethoprim | 5.0 | 5.4 | 3.0 | 0 | 5.1 |
| Ciprofloxacin | 2.0 | 2.1 | 3.0 | 0 | 2.1 |
| Gentamicin | 1.0 | 1.5 | 0 | 7.1 | 1.4 |
| Ampicillin | 1.0 | 1.5 | 0 | 0 | 1.3 |
| Antiprotozoals | | | | | |
| Sulfadoxine/pyrimethymine | 10.0 | 14.5 | 24.2 | 14.3 | 14.3 |
| Quinine | 5.0 | 2.9 | 6.1 | 0 | 3.3 |
| Metronidazole | 3.0 | 2.5 | 3.0 | 0 | 2.5 |
| Artesunate | 0 | 1.2 | 0 | 7.1 | 1.1 |
| Anthelmintics | | | | | |
| Mebendazole | 57.0 | 53.3 | 51.5 | 50.0 | 53.7 |
| Levamisole | 8.0 | 12.7 | 18.2 | 14.3 | 12.2 |
| Scabicides | | | | | |
| Benzyl benzoate | 5.0 | 5.2 | 0 | 0 | 4.8 |
| Analgesics, antipyretics | | | | | |
| Paracetamol | 33.0 | 33.4 | 48.5 | 50.0 | 34.5 |
| Metamizole | 30.0 | 23.2 | 39.4 | 35.7 | 25.4 |
| Acetylsalicylic acid | 12.0 | 6.0 | 3.0 | 0 | 6.7 |
| Anti-inflammatory medicines | | | | | |
| Diclofenac | 0 | 2.9 | 3.0 | 0 | 2.4 |
| Antihistamines | | | | | |
| Promethazine | 4.0 | 6.4 | 6.1 | 0 | 5.9 |
| Chlorphenamine | 1.0 | 2.5 | 0 | 0 | 2.1 |
| Dexchlorpheniramine | 0 | 2.3 | 0 | 0 | 1.7 |
| Corticosteroids | | | | | |
| Dexamethasone | 6.0 | 5.4 | 0 | 7.1 | 5.2 |
| Hydrocortisone | 2.0 | 4.4 | 6.1 | 7.1 | 4.1 |
| Cardiovascular medicines | | | | | |
| Furosemide | 1.0 | 2.9 | 0 | 0 | 2.4 |
| Gastrointestinal medicines | | | | | |
| Metoclopramide | 6.0 | 10.2 | 12.1 | 7.1 | 9.5 |

*(Continued)*

**Table 4.** (Continued)

| Percentage of patients treated with (%) | Children 0–11 years | Adolescents & adults > 11 years | Breast-feeding women | Pregnant women | All patients |
|---|---|---|---|---|---|
| | N = 100 | N = 482* | N = 33 | N = 14 | N = 629 |
| Oral rehydration solution | 7.0 | 5.4 | 3.0 | 7.1 | 5.6 |
| Aluminium Hydroxide | 0 | 3.1 | 3.0 | 0 | 2.5 |
| Papaverine | 1.0 | 2.3 | 3.0 | 21.4 | 2.5 |
| Nervous system medicines | | | | | |
| Diazepam | 14.0 | 18.5 | 15.2 | 0 | 17.2 |
| Phenobarbital | 1.0 | 4.8 | 0 | 7.1 | 4.0 |
| Chlorpromazine | 2.0 | 3.9 | 9.1 | 0 | 3.8 |
| Vitamins | | | | | |
| Multivitamins | 5.0 | 8.5 | 6.1 | 21.4 | 8.1 |
| Vitamin B6 complex | 3.0 | 5.8 | 6.1 | 7.1 | 5.4 |
| Perfusion solutions | | | | | |
| Hypertonic glucose | 7.0 | 7.5 | 9.1 | 14.3 | 7.6 |
| NaCl | 0 | 4.1 | 0 | 14.3 | 3.5 |
| Ringer lactate | 1.0 | 2.9 | 3.0 | 7.1 | 2.7 |
| Glucose | 2.0 | 1.9 | 6.1 | 0 | 2.1 |

*All adolescents and adults except breastfeeding / pregnant women

availability have not been collected at the time of the NECT-FIELD study. Therefore we cannot conclude on the relative importance of how lack of resources, supply chain, availability, and access influenced the utilization of concomitant drugs in our setting.

Another limitation of this present analysis is the replication and generalisability of the results, as we are only relying on the prescription habits of the clinicians and clinical teams of our six study sites. In our knowledge, this is the first study reporting the real-life prescription of concomitant medicine during second stage sleeping sickness treatment. In order to reassure our findings we compared the AEs reported in recent clinical trial [10, 16] and field cohorts [19] by compiling together the safety profiles in Table 5. Results suggests that the comorbidity and AE profile of

**Table 5. Overview of adverse event profile during HAT therapy in published clinical trials.**

| Percentage of patients with (%) | NECT vs DFMO Phase III trial [16] *,# | | NECT MSF [19] # | Fexinidazole vs NECT Phase III trial [10] | |
|---|---|---|---|---|---|
| | NECT N = 143 | DFMO N = 143 | NECT N = 684 | Fexinidazole N = 264 | NECT N = 130 |
| Gastrointestinal disorders | | | | | |
| Vomiting | 48$ | 20$ | 39 | 28 | 29 |
| Nausea | $ | $ | 8 | 26 | 19 |
| Diarrhoea | 6 | 29 | 7 | 3 | 4 |
| Abdominal pain | 25 | 29 | 41 | 10 | 12 |
| General disorders and administration site condition | | | | | |
| Fever | 26 | 43 | - | 9 | 19 |
| Asthenia | 24 | 20 | - | 23 | 14 |
| Fatigue | - | - | 11 | - | - |
| Injection site disorders | 10 | 11 | - | - | - |

(Continued)

**Table 5.** (Continued)

| Percentage of patients with (%) | NECT vs DFMO Phase III trial [16] *,# | | NECT MSF [19] # | Fexinidazole vs NECT Phase III trial [10] | |
|---|---|---|---|---|---|
| | NECT N = 143 | DFMO N = 143 | NECT N = 684 | Fexinidazole N = 264 | NECT N = 130 |
| Nervous system disorders | | | | | |
| Headache | 39 | 46 | 30 | 35 | 24 |
| Dizziness | 18 | 17 | 20 | 19 | 13 |
| Convulsions | 13 | 9 | 4 | 2 | 8 |
| Tremor | 6 | 1 | 7 | 22 | 11 |
| Coma or loss of consciousness | 1 | 2 | - | - | - |
| Metabolism and nutrition disorders | | | | | |
| Anorexia or decreased appetite | 25 | 14 | 21 | 21 | 19 |
| Psychiatric disorders | | | | | |
| Insomnia | 10 | 10 | 12 | 28 | 12 |
| Agitation | 3 | 8 | - | 4 | 1 |
| Mood disorders, anxiety, or depression | 1 | 1 | 8 | 4 | - |
| Mental confusion | 4 | 1 | - | - | - |
| Hallucinations, delirium, or psychosis | 1 | 1 | 3 | 3 | 3 |
| Musculoskeletal and connective tissue disorders | | | | | |
| Musculoskeletal pain | 30 | 30 | 25 | - | - |
| Back pain | - | - | - | 11 | 9 |
| Neck pain | - | - | - | 9 | 5 |
| Skin and sub-cutaneous tissue disorders | | | | | |
| Pruritus or cutaneous pruritus | 9 | 19 | 4 | 4 | 3 |
| Dermatitis or skin rash | 3 | 14 | - | - | - |
| Vascular disorders | | | | | |
| Hypotension | 4 | 3 | - | - | - |
| Hypertension | 4 | 13 | - | 5 | 1 |
| Cardiac disorders | | | | | |
| Palpitation or arrhythmia | 19 | 22 | - | 5 | 4 |
| Infections and infestations | | | | | |
| All kind of infections | 10 | 18 | - | 8 | 6 |
| Blood and lymphatic system disorders | | | | | |
| Anaemia | - | - | - | 9 | 11 |

*Only treatment related adverse event are reported in this trial (Priotto 2009) [16]

#Adverse event recorded in NECT phase III trial [16] and in NECT MSF [19] have not been coded with MedDRA dictionary

$In NECT phase III [16], nausea and vomiting were reported together and not differentiated

"-"AE term not reported in the respective publication

patients were comparable among all these studies and that concomitant medication needs would be more or less be similar among the different HAT treatment options and treatment centers.

In order to better manage the clinical profile of second stage HAT patients and the AEs of NECT, we propose a list of minimal essential drugs based on the results of this analysis and on the WHO EML that should be available at no cost to the patient night and day in treatment wards (Table 6) [20]. This recommendation provides new evidence to programs, health systems, donors, and other actors to plan, fund, and supply the essential medicines adapted to treat second stage sleeping sickness patients.

**Table 6. Recommended minimal essential medication to be delivered at a HAT treatment centre to treat main HAT symptoms, frequent comorbidities, and adverse events.**

| | |
|---|---|
| Antibacterials | Corticosteroids |
| Amoxicillin, *po* | Dexamethasone, *iv* |
| Ampicillin, *iv* | Hydrocortisone, *iv* |
| Ceftriaxone, *iv* | |
| Cotrimoxazole, *po* | Cardiovascular medicines |
| Ciprofloxacin, *po and iv* | Furosemide, *po* |
| Gentamycin, *iv* | Atenolol, *po* |
| Antiprotozoals | Gastrointestinal medicines |
| Metronidazole, *po* | Metoclopramide, *po* |
| Artesunate amodiaquine, *po* | Omeprazole, *po* |
| Artesunate lumefantrine, *po* | Oral rehydration salts, *po* |
| Artesunate, *im or iv* | |
| | Nervous system medicines |
| Anthelmintics | Chlorpromazine, *po* |
| Mebendazole or levamisole, *po* | Haloperidol, *po* |
| | Diazepam, *po* |
| Analgesics, antipyretic | Phenobarbital, *iv* |
| Paracetamol, *po* | |
| | Vitamins |
| Anti-inflammatory medicines | Vitamin B6 complex, *po* |
| Ibuprofen, *po* | |
| | Perfusion Solutions |
| Antihistamines | Glucose 5%, *iv* |
| Chlorpheniramin, *po* | NaCl, *iv* |
| Promethazine, *po* | Ringer lactate, *iv* |

In a further step and in order to improve and generalize the proposed list, analyses of the concomitant drugs used during fexinidazole [10] and acoziborole (registered under clinical-trials.gov: NCT03087955) trials for second stage sleeping sickness patients shall be performed in a near future.

## Acknowledgments

We thank the patients and their families for their participation in the NECT-FIELD study, and the medical and support staff of the Ministry of Health hospitals and the national HAT control programme for their patient care. The World Health Organisation (WHO) provided the drugs for free and supported logistics together with Médecins sans Frontières (MSF). The HAT Platform in Kinshasa provided liaison with the local ethics committee and the local authorities. The Swiss Tropical and Public Health Institute supported the implementation and monitoring of the NECT-FIELD study. Data management was provided by RCTs, Lyon, France.

## Author Contributions

**Conceptualization:** Andrea Kuemmerle, Caecilia Schmid, Olaf Valverde Mordt.

**Formal analysis:** Andrea Kuemmerle.

**Methodology:** Andrea Kuemmerle, Olaf Valverde Mordt.

**Project administration:** Andrea Kuemmerle.

**Resources:** Olaf Valverde Mordt.

**Supervision:** Olaf Valverde Mordt.

**Validation:** Caecilia Schmid, Victor Kande, Wilfried Mutombo, Medard Ilunga, Ismael Lumpungu, Sylvain Mutanda, Pathou Nganzobo, Digas Ngolo, Mays Kisala, Olaf Valverde Mordt.

**Visualization:** Andrea Kuemmerle, Caecilia Schmid, Olaf Valverde Mordt.

**Writing – original draft:** Andrea Kuemmerle, Caecilia Schmid, Olaf Valverde Mordt.

**Writing – review & editing:** Victor Kande, Wilfried Mutombo, Medard Ilunga, Ismael Lumpungu, Sylvain Mutanda, Pathou Nganzobo, Digas Ngolo, Mays Kisala.

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
