## [Decision Letter · Decision Letter 0]

28 Oct 2019

Dear Dr. Kuemmerle:

Thank you very much for submitting your manuscript "Prescription of concomitant medications in patients treated with Nifurtimox Eflornithine Combination Therapy (NECT) for second stage sleeping sickness in the Democratic Republic of the Congo" (#PNTD-D-19-01387) for review by PLOS Neglected Tropical Diseases. Your manuscript was fully evaluated at the editorial level and by independent peer reviewers. The reviewers appreciated the attention to an important problem, but raised some substantial concerns about the manuscript as it currently stands. These issues must be addressed before we would be willing to consider a revised version of your study. We cannot, of course, promise publication at that time.

We therefore ask you to modify the manuscript according to the review recommendations before we can consider your manuscript for acceptance. Your revisions should address the specific points made by each reviewer. 

When you are ready to resubmit, please be prepared to upload the following:

(1) A letter containing a detailed list of your responses to the review comments and a description of the changes you have made in the manuscript.

(2) Two versions of the manuscript: one with either highlights or tracked changes denoting where the text has been changed (uploaded as a "Revised Article with Changes Highlighted" file); the other a clean version (uploaded as the article file).

(3) If available, a striking still image (a new image if one is available or an existing one from within your manuscript). If your manuscript is accepted for publication, this image may be featured on our website. Images should ideally be high resolution, eye-catching, single panel images; where one is available, please use 'add file' at the time of resubmission and select 'striking image' as the file type. 

Please provide a short caption, including credits, uploaded as a separate "Other" file. If your image is from someone other than yourself, please ensure that the artist has read and agreed to the terms and conditions of the Creative Commons Attribution License at http://journals.plos.org/plosntds/s/content-license (NOTE: we cannot publish copyrighted images). 

(4) If applicable, we encourage you to add a list of accession numbers/ID numbers for genes and proteins mentioned in the text (these should be listed as a paragraph at the end of the manuscript). You can supply accession numbers for any database, so long as the database is publicly accessible and stable. Examples include LocusLink and SwissProt.

(5) To enhance the reproducibility of your results, we recommend that you deposit your laboratory protocols in protocols.io, where a protocol can be assigned its own identifier (DOI) such that it can be cited independently in the future. For instructions see http://journals.plos.org/plosntds/s/submission-guidelines#loc-methods

While revising your submission, please upload your figure files to the Preflight Analysis and Conversion Engine (PACE) digital diagnostic tool, https://pacev2.apexcovantage.com/ PACE helps ensure that figures meet PLOS requirements. To use PACE, you must first register as a user. Then, login and navigate to the UPLOAD tab, where you will find detailed instructions on how to use the tool. If you encounter any issues or have any questions when using PACE, please email us at figures@plos.org.

We hope to receive your revised manuscript by Dec 27 2019 11:59PM. If you anticipate any delay in its return, we ask that you let us know the expected resubmission date by replying to this email.

To submit a revision, go to https://www.editorialmanager.com/pntd/ and log in as an Author. You will see a menu item call Submission Needing Revision. You will find your submission record there. 

Sincerely,

Alvaro Acosta-Serrano

Deputy Editor

Alvaro Acosta-Serrano

Deputy Editor

The results in this paper are part of the NECT-FIELD study (https://clinicaltrials.gov/ct2/show/NCT00906880). The main outcome of this study (the proportion of patients discharged alive from hospital) was publised in 2012 (https://www.ncbi.nlm.nih.gov/pubmed/23209861), in that publication, submitted as additional information, the authors stated "Twenty-four months after end of treatment, patients will be assessed for the final effectiveness of treatment; these results will be reported later". Effectiveness is indeed one of the secondary outcomes of the NECT-FIELD study as indicated in the clinicaltrial.gov website: Effectiveness: The clinical cure rate (Survival without clinical and/or parasitological signs of HAT) [ Time Frame: 24 months after treatment ]. As the effectiveness results are related to the data presented in the manuscript submitted to PLoS NTDs. We would appreciate if the authors could clarify if the effectiveness results of the NECT-FIELD study have been or will be published.

Reviewer's Responses to Questions

**Key Review Criteria Required for Acceptance?**

**Methods**

-Are the objectives of the study clearly articulated with a clear testable hypothesis stated?

-Is the study design appropriate to address the stated objectives?

-Is the population clearly described and appropriate for the hypothesis being tested?

-Is the sample size sufficient to ensure adequate power to address the hypothesis being tested?

-Were correct statistical analysis used to support conclusions?

-Are there concerns about ethical or regulatory requirements being met?

Reviewer #1: Methods are very simple, just a data summary, all from one clinical trial. But in order to support some of their conclusions, they could add some data from routine practice outside clinical trials, some data on cost, etc, to allow comparisons that they insinuate in a superficial way.

Reviewer #2: The objectives of the study are well stated. The study design is appropriate for the stated objectives. Study population has been well described in the methodology and the sample size is sufficient to power this study. Descriptive statistics were used and this suffices for the kind of study objectives that the authors set out to determine. Being review of secondary data from a clinical trial, ethical concerns were adequately addressed.

Reviewer #3: The objective of the study are clear : indication of the use of concomitant medication during HAT treatment. The study design is appropriate and the population clearly described. The sample size is adequate and correct statistical analysis provided. Ethical comitees in both countries have been consulted.

**Results**

-Does the analysis presented match the analysis plan?

-Are the results clearly and completely presented?

-Are the figures (Tables, Images) of sufficient quality for clarity?

Reviewer #1: Results are clearly presented, although in a minimalist way. 

From that clinical trial, the results of most relevance for public health (the effectiveness of NECT), are not presented.

Reviewer #2: The results presented match the analysis plan. Results were presented in an exhaustive and logical manner, from baseline characteristics to the main findings in line with the objectives.The tables are clear and well structured (columns and rows are relevant).

Reviewer #3: The analysis is correct and the results clear. The tables are complexe but complete.

**Conclusions**

-Are the conclusions supported by the data presented?

-Are the limitations of analysis clearly described?

-Do the authors discuss how these data can be helpful to advance our understanding of the topic under study?

-Is public health relevance addressed?

Reviewer #1: Several conclusions are not supported by the data presented.

Reviewer #2: The authors have described the most common types of concomitant medications prescribed in patients with late stage sleeping sickness. These include analgesics/antipyretics, antimalarial and anti-helminthic drugs. They further go ahead to suggest a list of medicines that should be availed alongside NECT. 

The resource-limited setting of occurrence and treatment of sleeping sickness implies that preparations for adverse event management is defective. Various treatment centres are likely to have varying stock levels for the concomitant medicines that they may need at a certain point. The suggestions from authors contributes to the standardization of the medicines list.

These claims are novel and demonstrates the authors’ deep understanding of the gaps and therefore the needs at hand.

The context of the findings is also in line with the rest of the other sections of the manuscript. 

The paper is outstanding in the field of treatment for sleeping sickness. The disease presentation in itself often warrants concomitant medicines because of the severity of some presenting symptoms. Similarly, some adverse events experienced during the course of treatment can be life threatening. The authors were able to differentiate these two time points. This gives insight into the requirements before and during treatment.

Reviewer #3: The conclusions are clear and the limitations shortly described. The results should help to implement concomitant medication for the treatment of HAT.

**Editorial and Data Presentation Modifications?**

Reviewer #1: (No Response)

Reviewer #2: 1. Lines 101 and 102 reads: ‘Prior to NECT, the patients received standard pre-treatment according to the national guidelines, consisting of antimalarial, anthelminthic, and, if required, antipyretic/analgesic medication.’ This gives an impression that all patients receive antimalarial as a pre-requisite. Please confirm this.

This contradicts line 37/38 where anti-malarial medicine is classified as ‘additional’.

2. Line 40 and 41 reads ‘Treatment of encephalopathic syndrome or similar neurological disorders was used in a relevant proportion of patients (approximately 5%)’. The word ‘relevant’ seems out of place. Did you mean ‘significant’?

3. Line 44 reads ‘…even if some prescription habits would need to be adapted in the future’. it is not well aligned to the statement preceding it. Gives a different meaning.

4. The title should specify that participants had T.b gambiense.

Reviewer #3: Line 96 : should be "the NECT-FIELD study" instead of NECT-FIELD.

Line 112 : not really clear the sentence " AES in more than 5% of the patients and concomitant medications in more than 2.5%" ? Could it be clarified ?

Line 136 and table 2 : "such as neurological, sleeping and..." the sentence neurological does not mean anything here : has to be removed also I do not understand why there are no other neurological signs described at least abnormal movements and walking disorders which are characteristic of advanced stage.

Table 3 : not clear what SOC and LLT means and makes in the first line of the table ?

**Summary and General Comments**

Reviewer #1: General comments: 

This paper provides a description of the concomitant medication given to patients included in a trial, known as NECT-FIELD, evaluating the NECT treatment for second stage sleeping sickness.

The first and principal comment is that the most important results of that trial, the efficacy of NECT in the real-life setting, in children, in pregnant and lactating women, were not published, and are not available anywhere. These data are much more awaited by the scientific and public health community, than these data on concomitant medication.

The trial was successfully completed several years ago, including the post-treatment follow-up, and the effectiveness data should be published, taking priority to the data presented in this paper. 

Now regarding this article, the results presented are simply an enumeration of medicines given and their frequency. Only in the discussion the authors make some critique and suggest possible improvements, but without explaining their critique nor providing any data supporting the changes proposed. 

Specific comments:

Line 77: The EMA’s opinion did not recommend fexinidazole for all second stage patients. They excluded “stage 2 patients with >100 WBC per μL of CSF”. The authors should use the appropriate references for this, such as the EMA report available in their website, or the article published by EMA authors in PLOS NTD: Pelfrene et al. The European Medicines Agency's scientific opinion on oral fexinidazole for human African trypanosomiasis. PLoS Negl Trop Dis. 2019 Jun 27;13(6):e0007381

Lines 80-83: For a complete presentation of the NECT-FIELD study, the authors should not forget to mention that efficacy was also documented, but never published.

Line 96: Instead of “design, methods, and results”, it should be “design, methods, and safety results” (because efficacy results were not included in the Schmid paper).

Line 103: As these patients were all enrolled in a clinical trial, is it true that “Concomitant medication was prescribed according to the decision of the investigator and the local guidelines.”? Didn’t the study protocol impose any concomitant medication? This is important in the interpretation of results. How much these results represent the practice in DRC (or elsewhere) outside clinical trials?

Lines 181-188: It’s not clear why the authors focus on the encephalopathic syndrome linked to melarsoprol, and they seem to judge the appropriateness of use of certain drugs related to this syndrome, which does not even appear among the AEs in the study (and none of the patients received melarsoprol). It would be logical that they assess the use of drugs in relation to the AEs observed.

Lines 193-195: The authors say that using metamizol and papaverine is inadequate. They don’t explain why, except that they are “banned in many countries with stringent drug regulation”. This argumentation is way too superficial to be acceptable. Just by searching a little I can see that metamizole is banned only in a few countries (UK, Sweden, USA, Canada) because of agranulocytosis observed in people from northern Europe, or having population from that origin. Other countries use it widely and don’t have the same problem. India banned it at some point but they lifted the ban. The only argument offered by the authors would then be that metamizole should not be used in Africa because it provokes problems in northern Europeans (!). But perhaps the authors have other data that they can share, to support their recommendations. There are at least 2 recent review papers on metamizole published, which have much different conclusions: one says “For short-term use in the hospital setting, metamizole seems to be a safe choice when compared to other widely used analgesics.” I didn’t spend time searching on papaverine, but the authors should.

Line 195: Perhaps the statement “inadequate choices in the drug supply of the centres were mainly driven by the scarce financial resources available”, could be supported with some concrete examples giving the cost of choices made, versus the cost of the alternatives.

Reviewer #2: In summary, there are no copyright issues to be concerned about. The manuscript has been submitted for consideration in line with earlier work done by the same authors. 

Additionally, ethical principles have been followed as by virtue of the details in the section with the same title and the methodology.

Clarifications based on the comments in the above.

Reviewer #3: The paper provides an opportunity to clarify what kind of concomitant medication should be available in HAT treating centers. As the number of HAT cases is low ,all additional data from well conducted trials are of great interest for the general use and are thus interesting to show to the scientific community and policy makers.

PLOS authors have the option to publish the peer review history of their article (what does this mean?). If published, this will include your full peer review and any attached files.

Reviewer #1: No

Reviewer #2: Yes: Andrew Edielu

Reviewer #3: Yes: BISSER SYLVIE

---

## [Decision Letter · Decision Letter 1]

3 Jan 2020

Dear Dr. Kuemmerle,

We are pleased to inform you that your manuscript, "Prescription of concomitant medications in patients treated with Nifurtimox Eflornithine Combination Therapy (NECT) for T.b. gambiense second stage sleeping sickness in the Democratic Republic of the Congo", has been editorially accepted for publication at PLOS Neglected Tropical Diseases.

Before your manuscript can be formally accepted and sent to production you will need to complete our formatting changes, which you will receive in a follow up email. Please note: your manuscript will not be scheduled for publication until you have made the required changes.

IMPORTANT NOTES

* Copyediting and Author Proofs: To ensure prompt publication, your manuscript will NOT be subject to detailed copyediting and you will NOT receive a typeset proof for review. The corresponding author will have one final opportunity to correct any errors when sent the requests mentioned above. Please review this version of your manuscript for any errors.

* If you or your institution will be preparing press materials for this manuscript, please inform our press team in advance at plosntds@plos.org. If you need to know your paper's publication date for media purposes, you must coordinate with our press team, and your manuscript will remain under a strict press embargo until the publication date and time. PLOS NTDs may choose to issue a press release for your article. If there is anything that the journal should know, please get in touch.

*Now that your manuscript has been provisionally accepted, please log into EM and update your profile. Go to http://www.editorialmanager.com/pntd, log in, and click on the "Update My Information" link at the top of the page. Please update your user information to ensure an efficient production and billing process.

*Note to LaTeX users only - Our staff will ask you to upload a TEX file in addition to the PDF before the paper can be sent to typesetting, so please carefully review our Latex Guidelines [http://www.plosntds.org/static/latexGuidelines.action] in the meantime.

Best regards,

Albert Picado

Associate Editor

Alvaro Acosta-Serrano

Deputy Editor

Reviewer's Responses to Questions

**Key Review Criteria Required for Acceptance?**

**Methods**

-Are the objectives of the study clearly articulated with a clear testable hypothesis stated?

-Is the study design appropriate to address the stated objectives?

-Is the population clearly described and appropriate for the hypothesis being tested?

-Is the sample size sufficient to ensure adequate power to address the hypothesis being tested?

-Were correct statistical analysis used to support conclusions?

-Are there concerns about ethical or regulatory requirements being met?

Reviewer #1: Methods are appropriately described and there are no particular concerns.

**Results**

-Does the analysis presented match the analysis plan?

-Are the results clearly and completely presented?

-Are the figures (Tables, Images) of sufficient quality for clarity?

Reviewer #1: Results are well presented.

**Conclusions**

-Are the conclusions supported by the data presented?

-Are the limitations of analysis clearly described?

-Do the authors discuss how these data can be helpful to advance our understanding of the topic under study?

-Is public health relevance addressed?

Reviewer #1: Conclusions are supported by the data, limitations described, public health relevance addressed.

**Editorial and Data Presentation Modifications?**

Reviewer #1: No particular comments.

**Summary and General Comments**

Reviewer #1: The authors have addressed the issues raised by the review and the paper has improved considerably.

PLOS authors have the option to publish the peer review history of their article (what does this mean?). If published, this will include your full peer review and any attached files.

Reviewer #1: No

---

## [Editor Report · Acceptance letter]

22 Jan 2020

Dear Dr. Kuemmerle,

We are delighted to inform you that your manuscript, "Prescription of concomitant medications in patients treated with Nifurtimox Eflornithine Combination Therapy (NECT) for *T.b. gambiense* second stage sleeping sickness in the Democratic Republic of the Congo," has been formally accepted for publication in PLOS Neglected Tropical Diseases.

Best regards,

Serap Aksoy

Editor-in-Chief

Shaden Kamhawi

Editor-in-Chief
